# Wheat Germ and Lipid Oxidation: An Open Issue

**DOI:** 10.3390/foods11071032

**Published:** 2022-04-01

**Authors:** Silvia Marzocchi, Maria Fiorenza Caboni, Marcello Greco Miani, Federica Pasini

**Affiliations:** 1Department of Agricultural and Food Sciences and Technologies, University of Bologna, Piazza Goidanich 60, 47521 Cesena, Italy; maria.caboni@unibo.it (M.F.C.); federica.pasini5@unibo.it (F.P.); 2Casillo Next Gen Food S.r.l., Via Sant’Elia Z.I., 70033 Corato, Italy; marcello.miani@casillogroup.it

**Keywords:** wheat germ, stabilization, lipid oxidation, oil quality

## Abstract

Wheat germ (WG)’s shelf life after the milling process is incredibly short because of the presence of enzymes that aggravate the oxidation process; thus, stabilization is required in order to exploit the nutrients and bioactive compounds within WG. The critical point for the oxidation process is the mechanical treatment used to separate WG from the kernel, which exposes the lipid fraction to the air. Showing the connection between the quality of durum wheat, considering its storage management, and wheat germ oil (WGO), extracted with a cold press, solvent and supercritical CO_2_, is the aim of the study. The acidity and peroxide values were analyzed to evaluate lipid oxidation, while fatty acids, tocols, sterols and policosanols were evaluated for WGO characterization. The first fundamental step to control lipid oxidation is raw material management. Subsequently, the tempering phase of durum wheat, which is applied before the degermination process, is the most critical point for oxidation to develop because of the increase in moisture in the caryopsis and the activation of lipase and lipoxygenase. This represents a paradox: in order to stabilize the germ with degermination, first it seems inevitable to carry out a process that destabilizes it. To retains its highest quality, this will lead to a better use of the whole grain by reducing WG and by-product waste.

## 1. Introduction

*Triticum aestivum* is an hexaploid species that accounts for about 95% of wheat grown annually, and it is commonly used for bread production; on the other hand, durum wheat (*Triticum turgidum* ssp. *durum* Desf) is a tetraploid species most cultivated in the Mediterranean sea area due to the climate and conditions, and it is used mainly for pasta and couscous [1,2]. Durum wheat is an important source of bioactive compounds, with their relative health benefits, that are mainly contained in the grain bran and germ tissue [3]. Dietary fiber comprises carbohydrates (11.5–15.5% of dry wheat grain) that are able to reach the large intestine or colon [1]. The aleurone layer and embryo are a good source of micronutrients, such as selenium, iron and zinc, which are fundamental for the correct organism development during ingestion and for promoting gut and body health [3]. The most abundant metabolites in wheat grain are phenolic compounds, and bound phenolic acids in particular, that have antioxidant, anti-inflammatory and anti-carcinogenic properties [4].

The whole wheat kernel is composed of three structures: the endosperm (80–85% of dry wheat grain), which is formed of starch and proteins and surrounded by layers called “aleurone”; the bran (13–17%), which is formed of pericarp and testa, a hydrophobic tissue composed of lignin and lipidic compounds [5]; and the wheat germ.

In order to obtain the separation between flour and wheat germ (WG), caryopsis undergoes debranning and milling processes. The initial debranning is used for “covered” cereals; in fact, it removes the outer layers of caryopsis, allowing the recovery of intact kernels without causing damage to the endosperm region [6], producing some by-products. Debranning is followed by the common milling process of wheat, with which flour and co-products, such as coarse bran, fine bran and WG, are obtained [7].

Wheat germ represents 2–3% of the whole wheat kernel, and it is a precious milling co-product. Because of its concentration of high quality compounds, such as proteins, minerals, flavonoids, sterols, and vitamin E and B, WG is considered to be the most beneficial part of wheat grain; in fact, it has antioxidant, antihyperlipidemic, hypocholesterolemic and anticancer effects [8,9]. Meanwhile, the widespread use of WG is limited due to its rapid oxidation development because of the high presence of unsaturated fatty acids and hydrolytic and oxidative enzymes, such as lipoxygenase and lipase [10,11].

Removing WG during wheat milling is necessary to increase flour shelf life, but, on the other hand, WG shelf life is drastically shortened, and its stabilization is required in order to exploit its nutrients and bioactive compounds [12,13].

The WG separation process can be conducted with a direct and indirect approach: the former is represented by degermination, the latter by gradual separation, with an evolution of the milling process over the years leading up to the debranning process [9].

On the other hand, it is well known that WG stabilization can be conducted with different strategies, namely, physical, chemical and biological approaches. Briefly, physical technologies are represented by heat treatments, microwave, infrared and gamma radiation [9] and thermal/mechanical treatments. Heat treatments include steaming, which leads to the complete inactivation of lipase, while the lipoxygenase is inactivated to an extent of 80–92%, after 15 min treatment at 125–130 °C [12]; fluidization, which involves mass transfer between the material and hot air in the fluid bed, and which leads to complete lipase inactivation and 13.5% residual lipoxygenase [14]; and roasting, which is also used for stabilizing WG because it reduces moisture and enzyme activity [15]. Li and collaborators [16] found the optimal infrared irradiation effect to be at 90 °C for 20 min, obtaining residual lipase and lipoxygenase activity of about 19% for both enzymes; Jha et al. [17] stabilized WG through γ-irradiation, with a 31% inactivation of lipase. The chemical stabilization of WG can be caried out with enzyme denaturation via acidification using hydrochloric acid or acetic acid [18] or oil removal through organic solvents or supercritical CO_2_ extraction [9]. The supercritical CO_2_ extraction simplifies the oil refining process and eliminates the solvent distillation stage. In addition, CO_2_ is nontoxic, nonflammable, noncorrosive and recyclable [19]. Finally, biological stabilization consists of a fermentation of WG by lactic acid bacteria in order to obtain enzyme denaturation as a consequence of acidification [20].

The critical point for the oxidation process is the mechanical treatment used to separate the WG from the kernel, which involves a necessary tempering phase and, consequently, lipase and lipoxygenase activation.

To the best of our knowledge, there are no studies that consider what happens in this preliminary step; thus, the aim of this study is to demonstrate the connection between the quality of durum wheat and wheat germ oil (WGO). Additionally, we sought to find a way to use WG with the highest quality, allowing producers to use the whole grain and reduce waste and by-products. As reported above, during the debranning and milling processes, different by-products are obtained, so the characterization of their acidity parameters is fundamental for our aims, as is the lipid characterization of WGO, all of which can be used to evaluate the quality of the extracted oil. In addition, WGO was extracted with different technologies, including mechanical extraction, extraction with solvents and extraction with supercritical CO_2_, to evaluate if technology can affect its quality.

## 2. Materials and Methods

### 2.1. Tempering Phase of Wheat

Before the milling process, the raw durum wheat underwent a laboratory tempering process in water. Different treated water was used: water as control (C), water with 3% NaCl (W3%) and water with 5% NaCl (W5%) in order to see which has the least impact on the acidity of the wheat.

### 2.2. Samples

Winter durum wheat (*Triticum turgidum* ssp. *durum*) was supplied by an Italian mill, Molino Casillo S.p.A., and was used for wheat germ oil (WGO) extraction with different technologies. The durum wheat was sowed between the beginning of November and the first half of December, and the following thermal requirements were met: 2–3 °C for germination and tillering, 10 °C for rising, 15 °C for flowering and 20 °C for ripening. Durum wheat was harvested when it reached full ripening and its humidity was less than 13% (in the third week of May).

Two debranning by-products (DB1 and DB2) and one milling by-product (MB) were supplied by Molino Casillo S.p.A. A total of four samples for each by-product (DB1, DB2 and MB) were collected from four different plant systems.

Wheat germ for WGO was separated from durum wheat endosperm with the initial debranning and milling process, followed by sieving in order to obtain pure wheat germ.

### 2.3. Mechanical Extraction of WGO

The cold press extraction of WGO was performed using an SK60/1 press (Karl Strähle GmbH & Co., Dettingen unter Teck, Germany). Wheat germ was separated from durum wheat caryopsis and other impurities, such as metal pieces and stones, and wheat germ oil was extracted with a cold press (15 kg/h capacity).

### 2.4. Solvent Extraction of WGO

The standard protocol according to the AOAC Official Method [21] was used for the Soxhlet extraction of WGO; about 60 g of ground wheat germ was placed in a cellulose extraction thimble and the process was carried out for 2 h for complete extraction using a refluxing hexane. The residual solvent was evaporated using a rotary evaporator, Laborota 4001-efficient (Heidolph). Each extraction was performed twice.

### 2.5. Supercritical CO_2_ Extraction of WGO

The supercritical fluid extraction of about 1 kg ground wheat germ was carried out for 3 h at 380 bar and 55 °C under a constant CO_2_ flow rate of 20 kg/h.

### 2.6. Analytical Methods

#### 2.6.1. Moisture Content

Moisture content (%) was determined according to the European Standard Method UNI EN ISO 712:2010 [22]. Each determination was calculated twice.

#### 2.6.2. Lipid Determination of Durum Wheat and By-Products

Soxhlet extraction, according to the method of AOAC 920.39B [23], was conducted for lipid extraction in all samples. Each extraction was performed twice.

#### 2.6.3. Free Acidity Determination

Acidity was determined by means of volumetric titration according to the UNI EN ISO 660:2009 [22] standard method. Each determination was calculated twice for each lipid extraction (*n* = 4).

#### 2.6.4. Peroxide Value

The peroxide value was determined by means of volumetric titration according to UNI EN ISO 3960:2010 [22]. Each determination was calculated twice for each lipid extraction (*n* = 4).

### 2.7. Wheat Germ Oil Characterization

#### 2.7.1. Fatty Acid Composition

The fatty acid composition was determined according to ISO 12966-2:2017 +ISO 12966-4:2015 [22]. The analyses were carried out using a gas chromatograph (Shimadzu, Tokyo, Japan) equipped with a flame ionization detector (GC-FID), using a capillary column (CP-Sil 88-l = 100 m, 0.32 mm ID, film thickness 0.25 µm; Supelco, Bellefonte, PA, USA). Each determination was calculated twice.

#### 2.7.2. Tocopherol Composition

Tocopherols were evaluated according to ISO 9936:2016 [22]. A sample amount was diluted in hexane and injected in a HPLC system (Agilent 1200 series, Palo Alto, CA, USA) operating in direct phase with a silica column 4.6 mm ID × 250 mm length (Luna Hilic Phenomenex). Reference tocopherols (Sigma Aldrich, Milano, Italy) were used for calibration curve construction for quantification. Each analysis were performed twice.

#### 2.7.3. Unsaponifiable Matter

For the unsaponifiable matter, WGO samples extracted with different technologies were treated with alcoholic KOH solution, according to ISO 3956:2000 [22]. Each extraction was carried out twice.

#### 2.7.4. Sterols Composition

The procedure for sterols content and composition was performed according to NGD 71-1989+NGD 72-1989 [22] using a gas chromatograph (Shimadzu, Tokyo, Japan) equipped with a flame ionization detector with a CPSil 8CB (Supelco, Bellefonte, PA, USA) capillary column (l = 30 m, 0.32 mm ID, film thickness 0.25 μm). Each determination was calculated twice for each insaponifiable extraction (*n* = 4).

### 2.8. Statistical Analysis

A one-way analysis of variance (ANOVA, with Tukey’s honest significant difference multiple comparison) was evaluated using Statistica 8 software (2006, StatSoft, Tulsa, OK, USA). *p*-values lower than 0.05 were considered to be statistically significant.

## 3. Results and Discussion

### 3.1. Moisture, Lipid Content, Free Acidity and Peroxide Value

It is interesting to see that after the tempering phase, a fundamental step for the milling process, a significant acidity increase has been registered. In durum wheat before the tempering phase, the acidity was 9.5%, which then significantly (*p* < 0.05) increased after treatment. In fact, it increased by about 18–30 %; using different treated water, values of acidity reached 13.3, 13.8 and 11.6% using C, W3% and W5%, respectively. Therefore, this preliminary step before degermination is a critical point with regard to lipase and lipoxygenase activity, because, by adding water, the lipase cleaves triglycerides in fatty acids and the lipoxygenase catalyzes the oxidation of polyunsaturated fatty acids [24,25].

Table 1 shows the moisture content (%), lipid content (%), free acidity (%) and peroxide value (meqO_2_/kg of fat) determined during the milling process after the tempering phase. It is possible to see that moisture content does not change along the whole milling process; in fact, it is in a range between 13.3 and 13.9%.

Lipid content, instead, is affected by the milling process. In the intact durum wheat, lipid content was 2.5%, and it increased in the by-products after debranning, reaching a content of 4.4 and 6.4% in DB1 and DB2, respectively. This is due to the concentration of germ and aleurone in the by-products; in fact, is well known that oil in wheat is mainly concentrated in this part of the caryopsis [26,27]. Finally, the lipid content in the milling by-product was 4.1% after the process.

As regards the free acidity, this was 4.8% in the initial mixture of durum wheat, and during the debranning and milling process, it significantly increased, reaching values of 6.5, 5.3 and 7.0% in DB1, DB2 and MB, respectively. This can be due to the treatment conditions to which the wheat is exposed (such as temperature, heat, oil-water interface and water) [28].

The PV value trend is closely linked to the acidity one; in fact, in the initial durum wheat, it was 1.8 meqO_2_/kg of fat, and it increased significantly during the milling process, but without exceeding the legal limit (20 meqO_2_/kg of fat). After the first and second debranning, the PV reached a value of 5.2 and 3.9 meqO_2_/kg of fat, respectively, and in the final by-product, it reached the most significant and highest value (6.5 meqO_2_/kg of fat).

Considering Table 1, it is possible to see that acidity following the milling process decreases with respect to the one registered after the tempering phase.

Table 2 shows the yield, acidity and peroxide value of the WGO extracted from the separated germ with the three different technologies. The yield, calculated on a dry basis, and acidity were affected by the extraction technology, while the peroxide values were not. In fact, WGO extraction with a solvent registered a yield of 16%, significantly higher (*p* < 0.05) than the ones registered for mechanical and supercritical CO_2_ extraction, which were 6.6 and 6.4%, respectively. The WGO extracted with supercritical CO_2_ had a value of acidity of about 34%, significantly higher (*p* < 0.05) than the one obtained with mechanical (25.8%) and solvent (16%) extraction. Compared to the acidity registered during the milling process (Table 1), it is evident that it increased after the WGO extraction due to the technological process. The peroxide value was not affected by the extraction technology, and the results were in line with the ones registered following the milling process (Table 1).

### 3.2. Characterization of WGO Extracted with Different Technologies

Table 3 reports the fatty acid composition (%) of the WGO extracted with different technologies. Its composition was not affected by the extraction technology, and the results are in line with the literature [11,29], which reported linoleic acid (C18:2, ~53–58%) to be the major fatty acid in WGO, followed by oleic acid (C18:1, ~18–23%), palmitic acid (C16:0, ~13–17%) and linolenic acid (C18:3, ~3–6%).

Unsaponifiable matter was recorded to be at a significantly higher content in the WGO extracted with solvents and supercritical CO_2_, at 4.1 and 5%, respectively, than the one registered in the WGO extracted with mechanical extraction (3.6%). The extracting conditions could affect the extraction efficiency because of the mixture of polar and non-polar compounds, which characterize the unsaponifiable matter [29]. Unsaponifiable matter contains tocopherols, tocotrienols and phytosterols, whose compositions are shown below; they are an important constituent of vegetable oils due to their health benefits [30].

Tocol amounts in durum wheat reach approximatively 60 mg/100 g db according to the literature [26], and, of their composition (Table 4) in WGO, β-tocotrienol was the most preponderant (60–88%), and the extraction technology that affected it above all was supercritical CO_2_. In fact, the WGO extracted with supercritical CO_2_ showed a significantly (*p* < 0.05) higher percentage of β-tocotrienol (88%) than the WGO extracted with a cold press and with a solvent, which presented a percentage of about 60%. On the other hand, α-tocopherol, β-tocopherol and α-tocotrienol were present at a significantly (*p* < 0.05) higher percentage in the WGO extracted with a cold press (12, 5 and 21%, respectively) and solvent (15, 4 and 18%, respectively) than in the WGO extracted with supercritical CO_2_ (6.5, 2 and 2%, respectively). The others tocols were extracted in traces and did not show any significantly differences. In general, our results are in line with the literature [31,32,33,34]; the few differences identified can be due to the cultivar investigated and the cultivation technology.

It is well known that phytosterol consumption can reduce cardiovascular disease risk and blood LDL cholesterol levels [35]. The total sterol amount in durum wheat is in a range between 70 and 95 mg/100 g db [36]. Eight sterols (Table 5) were quantified in the WGO, with a preponderance of β-sitosterol (31–35%), followed by campestanol (15–18%), sitostanol (16–17%) and campesterol (12–13%). In general, WGO extracted with supercritical CO_2_ showed a significantly (*p* < 0.05) higher concentration of sterols, in particular of campesterol, stigmasterol and β-sitosterol. Campestanol, sitostanol and Δ_7_-avenasterol, instead, had a significantly (*p* < 0.05) higher concentration in the WGO extracted with a cold press and solvents. These results are in accordance with, or slightly lower than, the literature [29,30,31,32,33,34], but cultivar, the origin of the wheat germ and cultivation and environmental factors must be considered.

## 4. Conclusions

For the different wheat germ oil extraction technologies, differences were registered with regard to wheat germ oil yield, while its composition was not affected by the extraction. Raw material management is the first critical point for oil quality, considering free fatty acids and the oxidation level, which may depend on cultivation, storage and degermination conditions; the tempering phase of durum wheat that is applied before the degermination process is the most critical point for the development of lipid hydrolysis and oxidation. This is because the increase in the amount of moisture in the caryopsis causes the activation of lipase and lipoxygenase. This represents a paradox: in order to stabilize the germ with degermination, first, it seems inevitable to carry out a process that destabilizes it. Therefore, the whole stabilization process must take account of the effective quality of the germ at that moment and, at the same time, of the final quality of the flour and germ by-products obtained. Further studies are necessary and interesting in order to obtain WG with high quality after its separation from the caryopsis; in this way, is it possible to use to use the whole grain, reducing waste and by-products.

## Figures and Tables

**Table 1 foods-11-01032-t001:** Moisture content, lipid content, free acidity and peroxide value (PV) of the different milling by-products.

	Moisture Content (%)	Lipid Content (%)	Free Acidity (%)	PV (meqO_2_/kg of Fat)
**DW**	13.7 ± 0.4 a	2.5 ± 0.2 c	4.8 ± 0.0 b	1.8 ± 0.0 d
**DB1**	13.8 ± 0.3 a	4.4 ± 0.2 b	6.5 ± 1.2 a	5.2 ± 0.2 b
**DB2**	13.9 ± 0.1 a	6.4 ± 0.3 a	5.3 ± 0.4 a	3.9 ± 0.4 c
**MB**	13.3 ± 0.6 a	4.1 ± 0.5 b	7.0 ± 0.7 a	6.5 ± 0.6 a

Abbreviation: DW, durum wheat after tempering phase; DB1, by-products after first debranning; DB2, by-product after second debranning; MB, by-product after milling. Results of the analysis of variance with Tukey’s test are shown: *p* < 0.05; letters in the same column show significantly different values within each parameter.

**Table 2 foods-11-01032-t002:** Yield (dry basis), free acidity and peroxide value of WGO extracted with different technologies.

	Yield (%)	Free Acidity (%)	PV (meqO_2_/kg of Fat)
**Mechanical extraction**	6.6 ± 0.9 b	25.8 ± 1.1 b	3.5 ± 0.8 a
**Solvent extraction**	16.0 ± 2.1 a	16.0 ± 0.9 c	4.1 ± 0.7 a
**Supercritical CO_2_ extraction**	6.4 ± 0.4 b	34.0 ± 2.1 a	3.6 ± 0.6 a

Results of the analysis of variance with Tukey’s test are shown: *p* < 0.05; letters in the same column show significantly different values within each parameter.

**Table 3 foods-11-01032-t003:** Fatty acid composition (%) of WGO extracted with different technologies.

*Fatty Acid*	*Mechanical Extraction*	*Solvent Extraction*	*Supercritical CO_2_ Extraction*
C16:0	14.8 ± 1.4 ab	14.8 ± 0.9 b	17.1 ± 1.2 a
C18:0	1.3 ± 0.3 a	1.3 ± 0.4 a	1.4 ± 0.3 a
C18:1	20.3 ± 1.5 a	22.0 ± 2.3 a	21.0 ± 1.6 a
C18:2	56.7 ± 1.8 a	56.0 ± 2.0 a	53.5 ± 0.3 b
C18:3	4.6 ± 0.9 a	3.9 ± 0.5 a	4.0 ± 0.3 a

Results of the analysis of variance with Tukey’s test are shown: *p* < 0.05; letters in the same row show significantly different values within each fatty acid.

**Table 4 foods-11-01032-t004:** Tocol composition (%) of WGO extracted with different technologies.

*Tocol*	*Mechanical Extraction*	*Solvent Extraction*	*Supercritical CO_2_ Extraction*
α-Tocopherol	12.9 ± 2.3 a	15.3 ± 2.2 a	6.5 ± 0.9 b
β-Tocopherol	5.3 ± 0.4 a	4.4 ± 0.7 a	1.9 ± 0.3 b
γ-Tocopherol	0.4 ± 0.0 a	0.4 ± 0.0 a	0.5 ± 0.0 a
δ-Tocopherol	0.1 ± 0.0 a	0.2 ± 0.0 a	0.1 ± 0.0 a
α-Tocotrienol	21.1 ± 6.0 a	18.1 ± 2.3 a	2.3 ± 0.2 b
β-Tocotrienol	60.1 ± 8.2 b	61.1 ± 3.7 b	88.3 ± 3.6 a
γ-Tocotrienol	0.1 ± 0.0 a	0.2 ± 0.0 a	0.1 ± 0.0 a
δ-Tocotrienol	0.1 ± 0.0 a	0.2 ± 0.0 a	0.2 ± 0.0 a

Results of the analysis of variance with Tukey’s test are shown: *p* < 0.05; letters in the same row show significantly different values within each sterol.

**Table 5 foods-11-01032-t005:** Single sterol composition (%) of WGO extracted with different technologies.

*Sterol*	*Mechanical Extraction*	*Solvent Extraction*	*Supercritical CO_2_ Extraction*
Campesterol	12.6 ± 0.9 ab	12.5 ± 0.4 b	13.8 ± 0.7 a
Campestanol	17.6 ± 1.0 a	18.5 ± 0.6 a	15.2 ± 0.8 b
Stigmasterol	3.2 ± 0.5 b	3.3 ± 0.3 b	4.1 ± 0.2 a
β-Sitosterol	32.7 ± 2.0 ab	31.5 ± 1.5 b	35.3 ± 1.0 a
Sitostanol	17.1 ± 1.4 ab	18.8 ± 0.8 a	16.3 ± 0.9 b
Δ_5_-Avenasterol	7.4 ± 1.0 a	8.1 ± 0.7 a	8.0 ± 0.7 a
Δ_7_-Stigmasterol	1.2 ± 0.2 b	1.6 ± 0.3 ab	1.8 ± 0.1 a
Δ_7_-Avenasterol	2.1 ± 0.3 a	2.2 ± 0.2 a	1.3 ± 0.2 b

Results of the analysis of variance with Tukey’s test are shown: *p* < 0.05; lowercase letters in the same row show significantly different values within each sterol.

## Data Availability

Data is contained within the article.

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
