# Peer review of "Wheat Germ and Lipid Oxidation: An Open Issue"

_foods, 2022, doi:10.3390/foods11071032_

Round 1

Reviewer 1 Report

This research is about the extraction method of wheat germ and its component analysis.I don't see the innovation in this article, which only compared the composition of the wheat germ obtained by the three extraction methods .The authors did not even state why these specific extraction conditions were chosen. and the conclusion ‘This represents a paradox: in order to stabilize the germ with degermination, first it seems inevitable to carry out a process that destabilizes it. ‘’ was unclear. Moreover, which part of the article reflects “whole grain”?

Author Response

Response to Reviewer 1 Comments

Point 1: This research is about the extraction method of wheat germ and its component analysis. I don't see the innovation in this article, which only compared the composition of the wheat germ obtained by the three extraction methods .The authors did not even state why these specific extraction conditions were chosen. and the conclusion ‘This represents a paradox: in order to stabilize the germ with degermination, first it seems inevitable to carry out a process that destabilizes it. ‘’ was unclear. Moreover, which part of the article reflects “whole grain”?

Response 1: The extraction methods were chosen because cold press and CO2 represent potentially sustainable and green extraction methods rather than solvent extraction. The conclusion sentence means that the degermination is the key step in order to stabilize the wheat germ in order to obtain wheat germ oil, but prior a tempering phase, adding water, is necessary and it involves in lipase and lipoxygenase activation. By adding water lipase cleaves triglycerides in fatty acids and lipoxygenase catalyse the oxidation of polyunsaturated fatty acids. The “whole grain” in title is because if you find a way to use germ in the best conditions then allows you to use the whole grain.

Reviewer 2 Report

In my opinion, the introduction is very perfunctory, there is no background, no description of durum wheat, I propose to improve it.

The methodology lacks information on cultivation conditions, what varieties of T.durum were tested (spring or winter?).

This paragraf could have been described a little better, not so briefly.

In my opinion, the methodology is too abbreviated.

Author Response

Response to Reviewer 2 Comments

Point 1: In my opinion, the introduction is very perfunctory, there is no background, no description of durum wheat, I propose to improve it.

Response 1: authors take into account the reviewer comments and introduction has been improved.

Point 2: The methodology lacks information on cultivation conditions, what varieties of T.durum were tested (spring or winter?).

This paragraf could have been described a little better, not so briefly.

In my opinion, the methodology is too abbreviated.

Response 2: authors take into account the reviewer comments and the part of the methodology has been improved.

Reviewer 3 Report

The most nutritionally valuable part of a wheat grain is the wheat germ, which is anatomically the smallest part of the wheat kernel. In wheat milling industry, wheat germ is a byproduct that is susceptible to lipid oxidation. Therefore, a study showing the possibility of better utilization of wheat germ is very useful. However, in this paper there are methods that need to be clarified (with respect to the material for extraction), and thus the results obtained by using these methods.

Chapter 2. Materials and Methods: Please explain the reasons why you used differently prepared raw materials for the different extraction methods (you performed mechanical extraction on broken or damaged whole durum wheat caryopses; solvent extraction on ground wheat germ; supercritical CO2 extraction on ground durum wheat). Whole grain extraction allowed extraction of the lipids and fat-soluble components from the whole grain, rather than extraction of the germ oils only (as indicated in the titles of subsections 2.3. and 2.5.).

Chapter 3. Results and Discussion:

Please explain how you calculated that the acidity increases by 18-30% after treating durum wheat with differently treated water.

In the explanation of the results in Table 2, it states "... WGO extracted from separated germ using the three different technologies", which is not consistent with the description of the extraction process in subsections 2.3. and 2.5.

Please explain how you calculated the yield shown in Table 2. Actually, the yield of oil extracted only from the germ (described in subsection 2.4.) should be significantly higher (16.0%) than the oil yield extracted from the whole durum wheat grain (described in subsections 2.3. and 2.5.), regardless of the extraction method.

In the introduction to their paper (https://www.ncbi.nlm.nih.gov/pmc/articles/PMC6616852/), Narducci et al. have quoted a sentence from the book chapter "Kernel components of technological value" ("Durum Wheat Chemistry and Technology"), according to which 15% of lipids are distributed in the bran (especially in the aleurone layer) and about 20% in the endosperm of durum wheat. For these reasons, the methods used and the results and conclusions presented should be further clarified given the differences in the starting material (wheat germ only/whole durum wheat grains) used in the different extraction methods.

Please indicate if the results shown in the tables are expressed on a dry basis.

Author Response

Response to Reviewer 3 Comments

Point 1: Chapter 2. Materials and Methods: Please explain the reasons why you used differently prepared raw materials for the different extraction methods (you performed mechanical extraction on broken or damaged whole durum wheat caryopses; solvent extraction on ground wheat germ; supercritical CO2 extraction on ground durum wheat). Whole grain extraction allowed extraction of the lipids and fat-soluble components from the whole grain, rather than extraction of the germ oils only (as indicated in the titles of subsections 2.3. and 2.5.).

Response 1: authors take into account the reviewer comments and this part of Materials & Methods has been corrected.

Point 2: Chapter 3. Results and Discussion: Please explain how you calculated that the acidity increases by 18-30% after treating durum wheat with differently treated water.

Response 2: for the calculation of acidity increase was calculated the percentage difference between the acidity value before the tempering phase (9.5%) and after tempering phase (13.3, 13.8 and 11.6%); in this way the percentage increase was between 18 and 30%.

Point 3: In the explanation of the results in Table 2, it states "... WGO extracted from separated germ using the three different technologies", which is not consistent with the description of the extraction process in subsections 2.3. and 2.5.

Response 3: authors take into account the reviewer comments and the subsections 2.3 and 2.5 in Material and Methods section have been corrected.

Point 4: Please explain how you calculated the yield shown in Table 2. Actually, the yield of oil extracted only from the germ (described in subsection 2.4.) should be significantly higher (16.0%) than the oil yield extracted from the whole durum wheat grain (described in subsections 2.3. and 2.5.), regardless of the extraction method.

Response 4: authors take into account the reviewer comment and correct the Material and Methods section about this part and, consequently, the Results and Discussion section. Yield of oil was calculated on dry basis.

Point 5: In the introduction to their paper (https://www.ncbi.nlm.nih.gov/pmc/articles/PMC6616852/), Narducci et al. have quoted a sentence from the book chapter "Kernel components of technological value" ("Durum Wheat Chemistry and Technology"), according to which 15% of lipids are distributed in the bran (especially in the aleurone layer) and about 20% in the endosperm of durum wheat. For these reasons, the methods used and the results and conclusions presented should be further clarified given the differences in the starting material (wheat germ only/whole durum wheat grains) used in the different extraction methods.

Response 5: authors correct the starting material in Material and Methods section

Point 6: Please indicate if the results shown in the tables are expressed on a dry basis.

Response 6: authors take into account the reviewer comments and the expression of results were indicated in the text.

Round 2

Reviewer 1 Report

The author made some improvements,  but there are still some problems. The introduction part is still very weak. This part should focus on the whole grain and stabilization of wheat germ; and the innovation of this article is not clear?

  Author Response

Response to Reviewer 3 Comments – Round 2

Point 1. The author made some improvements,  but there are still some problems. The introduction part is still very weak. This part should focus on the whole grain and stabilization of wheat germ; and the innovation of this article is not clear?

Response 1: The parte on wheat germ stabilization has been improved and innovation of the article has been clarified.

Reviewer 3 Report

Thanks for the corrections, but there are still parts in the paper that are not adequately explained.

In Chapter 2, Material and Methods, you did not explain how and with what equipment you performed the process of separating wheat germ from durum wheat endosperm.

In addition, in Chapter 2, Material and Methods, you state that you used "winter durum wheat ... was used for wheat germ oil (WGO) extraction with different technologies." You have not clearly explained why you used milling by-products, i.e., you have not clearly explained the relationship between the wheat germ oil extraction results and the milling by-products chemical analysis results (except for the sentence at the end of the introduction and the last two sentences in Chapter 3.1. about acidity) and the relationship to storage management. You need to explain in more detail why you used milling by-products because not all potential readers are familiar with the milling process of wheat.

Also, in almost all parts of the paper (from the abstract to the conclusions), you need to clearly explain the relationship between the results of your research on wheat germ oil and the "whole grain" in the title.

You did not specify whether you used controls (C), water with 3% NaCl (W3%), or water with 5% NaCl (W5%) for the laboratory tempering process of durum wheat.

Please reconcile the heading of Table 2 (moisture content, lipid content) with the results shown in the table (yield, PV) and with the explanations in the text.

Author Response

Response to Reviewer 3 Comments – Round 2

Thanks for the corrections, but there are still parts in the paper that are not adequately explained.

Point 1. In Chapter 2, Material and Methods, you did not explain how and with what equipment you performed the process of separating wheat germ from durum wheat endosperm.

Response 1: Specification on wheat germ separation has been added in Materials and Methods.

Point 2. In addition, in Chapter 2, Material and Methods, you state that you used "winter durum wheat ... was used for wheat germ oil (WGO) extraction with different technologies." You have not clearly explained why you used milling by-products, i.e., you have not clearly explained the relationship between the wheat germ oil extraction results and the milling by-products chemical analysis results (except for the sentence at the end of the introduction and the last two sentences in Chapter 3.1. about acidity) and the relationship to storage management. You need to explain in more detail why you used milling by-products because not all potential readers are familiar with the milling process of wheat.

Response 2: authors take account reviewer comment and informations about by-products have been added.

Point 3. Also, in almost all parts of the paper (from the abstract to the conclusions), you need to clearly explain the relationship between the results of your research on wheat germ oil and the "whole grain" in the title.

Response 3: authors take account reviewer comment and the relationship with wheat germ oil and whole grain has been clarified throughout the paper.

Point 4. You did not specify whether you used controls (C), water with 3% NaCl (W3%), or water with 5% NaCl (W5%) for the laboratory tempering process of durum wheat.

Response 4: all the water (control and treated) were used for the tempering process.

Point 5. Please reconcile the heading of Table 2 (moisture content, lipid content) with the results shown in the table (yield, PV) and with the explanations in the text.

Response 5: authors take into account reviewer comment and heading of Table 2 has been reconciled.
